# Rapid Learning without Catastrophic Forgetting in the Morris Water Maze

## Abstract

Machine learning models typically struggle to swiftly adapt to novel tasks while maintaining proficiency on previously trained tasks. This contrasts starkly with animals, which demonstrate these capabilities easily. The differences between ML models and animals must stem from particular neural architectures and representations for memory and memory-policy interactions. We propose a new task that requires rapid and continual learning, the sequential Morris Water Maze (sWM). Drawing inspiration from biology, we show that 1) a content-addressable heteroassociative memory based on the entorhinal-hippocampal circuit with grid cells that retain knowledge across diverse environments, and 2) a spatially invariant convolutional network architecture for rapid adaptation across unfamiliar environments together perform rapid learning, good generalization, and continual learning without forgetting. Our model simultaneously outperforms ANN baselines from both the continual and few-shot learning contexts. It retains knowledge of past environments while rapidly acquiring the skills to navigate new ones, thereby addressing the seemingly opposing challenges of quick knowledge transfer and sustaining proficiency in previously learned tasks.

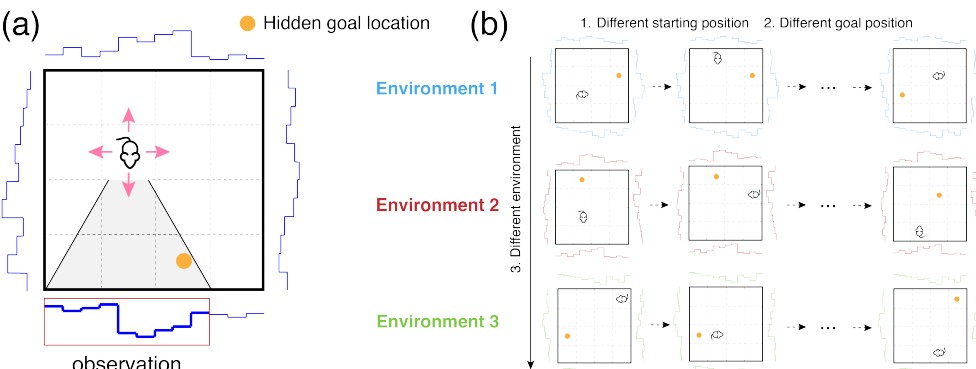

Figure 1: Schematic of the sequential Morris Water Maze task. **(a) The water maze environment.** The rodent icon represents the agent, arrows indicate the rodent's allowed actions, gold circles indicate the hidden platforms, and curves parallel to the walls of each environment denote patterns along the walls. The agent observes a portion of the wall pattern. **(b) our training setup.** Agents must generalize in three distinct ways. 1. Find a fixed goal location starting from random points in the environment. 2. Quickly learn new goal location within one environment, and reach it from random starting locations. 3. Learn various new environments, each with random start positions and multiple goal locations. The agent is evaluated on rapidly learning to navigate to new goal locations and in new environments and remembering navigation strategies from previously seen environments.

# 1 INTRODUCTION

Animals can *rapidly* learn new tasks that are conceptually similar to previously encountered tasks, but have different inputs and surface-level details; simultaneously, *they retain the ability to solve the previous tasks*. Neural modeling of this process of rapid conceptual knowledge transfer with retention of past learning has been limited. In some ways, rapid learning and learning retention seem to be in opposition: the former requires fast adaptation of parameters while the latter requires stable parameters. In machine learning, models tend to focus on either solving rapid learning and transfer, or on continual learning without forgetting; models tend not to do well at both.

Here, we build a biologically motivated neural model to solve a sequential version of the classic Morris Water Maze task (Morris, 1981; Vorhees & Williams, 2006), in which a rodent must find and then navigate to a submerged platform in a pool of cloudy water across multiple trials starting from different positions. We term our variant of this task the Sequential Morris Water Maze (sWM) task. This task necessitates sequential learning across multiple unique environments, each characterized by a different platform location. Within a single environment, the task demands two generalizations from the agent. First, it must generalize its learning from a variety of starting locations. Second, it must rapidly adapt to changes in the goal locations. In our sequential version of the task, an additional layer of complexity is introduced. Here, the agent is required to learn new environments while preserving the knowledge of the previous ones. This requirement tests the agent's ability to avoid catastrophic forgetting, a significant challenge in machine learning. Thus, the sWM task not only involves the aforementioned *intra-environment* generalization and adaptation but also *inter-environment* learning and memory retention.

Conventional unstructured neural networks suffer from catastrophic forgetting: a phenomenon where networks trained on a sequence of tasks fail to perform well on previously trained tasks (McCloskey & Cohen, 1989). Unstructured neural networks generally also lack an intrinsic ability to generalize rapidly to unseen tasks. Networks that perform rapid task transfer are typically extensively trained on a large number of related tasks (e.g. using multi-task learning techniques (Caruana, 1997) or meta-learning (Thrun & Pratt, 1998)).

We propose that certain inductive biases, like those present in the brain, allow networks to avoid these shortcomings and achieve performance on rapid and continual learning that is comparable to animals. It is known that animals use specialized computations in the hippocampus and entorhinal cortex to enable efficient spatial navigation and learning (O'Keefe & Dostrovsky, 1971; Hafting et al., 2005). We use a structured neocortical-entorhinal-hippocampal circuit, the Memory Scaffold with Heteroassociation (MESH) architecture (Sharma et al., 2022), to enable such generalization in the Water Maze *after training only on a single environment*. Our model proceeds as follows: first, MESH maps observation signals to a *grid cell* pattern, a type of spatial representation found in the entorhinal cortex. The grid code is then inputted into a randomly initialized, fixed Convolutional Neural Network (CNN), yielding a spatially invariant output feature vector. Lastly, this feature is processed by an attention module to determine the agent's action.

Our approach integrates a high-capacity content-addressable memory system with a spatially-invariant network specifically designed to facilitate zero-shot policy learning in new environments. Conceptually, this combination is beneficial as it allows the system to store and retrieve relevant information efficiently, while also adapting rapidly to new environments without requiring additional training. This functionality reflects the learning behavior of biological entities, contributing to the agent's capacity for both knowledge retention and rapid, flexible learning. We would like to emphasize that we are the first work that employs MESH in continual learning tasks.

The contribution of this paper is threefold:

- We propose a new lifelong learning task, sequential Morris Water Maze task (sWM), based on the widely used Morris Water Maze test of spatial learning in animals.

- We propose a neuro-inspired lifelong learning algorithm based on MESH (Sharma et al., 2022); the algorithm is specifically designed to perform rapid learning while retaining knowledge over long time-scales.

- In sWM, our method achieves significantly higher performance than baseline methods in both standard continual and few-shot learning.

## 2 RELATED WORK

### 2.1 CONTINUAL LEARNING IN ARTIFICIAL INTELLIGENCE

Continual learning methods can be categorized into three approaches; 1) regularization-based methods, 2) replay-based methods, and 3) architecture-based methods. Regularization-based methods (Cheung et al., 2019; Kirkpatrick et al., 2017; Li & Hoiem, 2017; Zenke et al., 2017) employ regularization terms to constrain the changes in model parameters to preserve previous model weights. They balance the trade-off between stability and plasticity in the learning process. EWC (Kirkpatrick et al., 2017) leverages the Fisher information matrix to estimate an importance matrix used for parameter regularization so that the network can remember old parameters. LwF (Li & Hoiem, 2017) finds the output logits from an old model trained on a previous task and distills them into a new model. Replay-based methods (Robins, 1995) prevent forgetting by forming a replay buffer, a small exemplar set of previous data, or synthetic data (Van de Ven et al., 2020) to interleave with new tasks during training. Since the memory size is constrained, there are several approaches to find smaller subset; reservoir sampling, reinforcement learning (Rebuffi et al., 2017), gradient-based selection (Aljundi et al., 2019). Another line of research employs existing sampling techniques and focuses on other aspects such as distillation (Douillard et al., 2020; Kang et al., 2022). Architecture-based methods focus on altering the model's *structure* to accommodate new tasks without affecting the performance of previous tasks. DEN (Yoon et al., 2018) dynamically expands neurons in the network. On the other hand, PNN (Rusu et al., 2016), DER (Yan et al., 2021) generates a new architectural backbone for each task, and FOSTER (Wang et al., 2022) distills a previous backbone network and a new backbone network into a single network applicable to the tasks corresponding to either backbone network.

### 2.2 CONTINUAL LEARNING IN NEUROSCIENCE

Unlike continual learning with an artificial neural network, biological neural networks do not suffer from catastrophic forgetting (Morris, 1981). Aimone et al. (2010) argue that adult-born neurons contribute to learning new information while separating previous patterns. In the Morris Water Maze task, where a rodent navigates toward a hidden escape platform relaying on distal cues, it directly heads to the platform even in an environment that was learned a few days ago (Morris, 1981; Vorhees & Williams, 2006). Place cells in the hippocampus play a key role in solving the task; they facilitate self-localization and route replay (Redish & Touretzky, 1998). Furthermore, they organize spatial information into separate maps when there is a significant shift in context or other non-spatial or spatial variables (*remapping*) (Colgin et al., 2008; Fyhn et al., 2007). This allows the rodent to remember each environment with associated platform location information, which enables it to find to navigate to the platform directly. Our method is based on MESH (Sharma et al., 2022) which models the entorhinal-hippocampal circuit.

## 3 MORRIS WATER MAZE TASK

We have developed a variant of the Morris Water Maze task called the sequential Morris Water Maze (sWM). This task assesses an artificial rodent's ability to remember previously explored environments while quickly learning new ones. In the original task, a rodent is placed in a circular tub filled with opaque fluid. Distal cues provide spatial information to the rodent. Inside the tank, there is a hidden platform that the rodent must find to avoid exhaustion from swimming. Once the rodent discovers the platform, it is placed in a different starting location within the same environment. This process is repeated multiple times. Then, the rodent is introduced to a different environment where the goal location and wall cues have changed, and the process repeats. Impressively, after training in multiple environments, the rodent retains knowledge of previous environments and rapidly navigates toward the platforms, even when faced with new ones.

For our task, we simplified the setup by using a square tub with distinctive markings on the walls as cues. These markings help the agent localize itself within the environment. The agent receives these cues as a vector input, which it uses to make informed navigation decisions. The agent's objective is to efficiently locate a hidden platform within the environment. The agent's movements are limited to four cardinal directions - north, south, east, or west.

Once the agent has been sufficiently trained in one environment, we introduce a sequential training regime. In this phase, the agent is exposed to both familiar and unfamiliar environments, with

Figure 2: Schematic of our model: MESH for Spatial Navigation (MSN) The agent observes a portion of wall. Observations, along with velocity inputs, are fed into a MESH network that produces grid cell activations representing the agent's location. An external memory module stores the grid code of the goal location. Grid codes of the current location and goal location are fed to the displacement network, a spatially-invariant convolutional neural network to produce a representation of the relative goal position. This is fed to a policy that produces actions.

different starting points in each. Varying the starting points adds complexity, and requires the agent to adapt its strategies based on its current position and the goal location.

Our task provides a comprehensive evaluation of the agent's cognitive abilities, specifically focusing on its capacity to retain knowledge from past experiences and its ability to quickly learn from new ones. These are qualities that biological entities, like rodents, naturally possess and demonstrate with remarkable efficiency. By replicating these attributes in our artificial agent, we aim to create a system capable of navigating complex tasks with similar adeptness.

## 4 MESH

MESH (Sharma et al., 2022) is a content-addressable memory (CAM) model based on the architecture of the neocortical-entorhinal-hippocampal memory circuit in the brain. Content-addressable memory models are networks that can store vectors (patterns to be memorized) as fixed points of their dynamics, and thereby recall/reconstruct them from noisy cues. Specifically, given a corrupted version of a previously encountered pattern, CAM models aim to reconstruct the original un-corrupted ground truth pattern. CAM models often suffer from a memory cliff problem: when the number of stored patterns crosses a certain threshold, the model not only fails to learn any new patterns, but also abruptly fails to recall all previously stored patterns. This is a form of *catastrophic forgetting*.

MESH addresses the memory cliff problem by constructing a fixed scaffold of pre-defined content-independent fixed points, which are then used to store the content-laden patterns through hetero-associative learning, thus mimicking the neocortical-entorhinal-hippocampal circuit to store patterns. The MESH architecture consists of three layers; features, hidden states, and labels, which biologically correspond to sensory input, place cell layer, and grid cell layer, respectively. We use grid code as labels instead of the $k$-hot labels in MESH. The place cell layer $\mathbf{p} \in \{-1, +1\}^{N_P}$ represents an $N_P$ dimensional binary vector, the grid cell layer $\mathbf{g} \in \{0, 1\}^{\sum \lambda_i}$ is defined as the concatenation of $\lambda_i$ dimensional one-hot vectors each of which represents a grid module in the brain, and the sensory layer is $N_s$ dimensional.

Before starting experiments, the memory scaffold (grid and place cells sates, as well as the projections between the grid and place cell layers) is *pre-defined*. The projection matrix from the grid cell layer to the place cell layer, $\mathbf{W}_{PG}$ is randomly generated so that it maintains an injective projection. On the other hand, the weight matrix from the place cell layer to the grid cell layer is trained by Hebbian learning such that it associates each active place cell (defining a place code) to the concurrently active grid cells (defining a corresponding grid code):

$$\mathbf{W}_{GP} = \frac{1}{|\mathbf{N}|} \sum_{\mu=1}^{\mu=N} \mathbf{g} \cdot (\mathsf{sign}(\mathbf{W}_{PG} \cdot \mathbf{g}))^T, \tag{1}$$

where $N$ is the number of training patterns.

When the agent explores the environment, the weights between sensory inputs and the place cells ($\mathbf{W}_{SP}$ and $\mathbf{W}_{PS}$) are learned by a pseudoinverse learning rule (Personnaz et al., 1985) in an online

manner (Tapson & van Schaik, 2013), yielding the following final weights:

$$\mathbf{W}_{SP} = \mathbf{S} \cdot \mathbf{P}^{\dagger}, \tag{2}$$

$$\mathbf{W}_{PS} = \mathbf{P} \cdot \mathbf{S}^{\dagger}, \tag{3}$$

where $\mathbf{S}$ and $\mathbf{P}$ are $N_s \times N$ and $Np \times N$ dimensional matrices of sensory patterns and place patterns respectively, and $\dagger$ indicates the pesudoinverse.

In summary, given the sensory input $\mathbf{s}_t$ at time $t$, the corresponding place cell and grid cell activations are computed through the model dynamics as follows:

$$\mathbf{p}_t = \mathsf{sign}(\mathbf{W}_{PS} \cdot \mathbf{s}_t), \tag{4}$$

$$\mathbf{g}_t = \mathsf{CAN}(\mathbf{W}_{GP} \cdot \mathbf{p}_t). \tag{5}$$

where $\mathsf{CAN}(\cdot)$ represents the continuous attractor recurrence in the grid layer that is implemented using a module-wise winner-take-all dynamics. This ensures that the equilibrium grid state is always a valid grid code i.e., a concatenation of one-hot vectors corresponding to each grid module.

The grid cell layer receives velocity signals (action input $\mathbf{a}_t$) for path integration, where the activated index for each grid cell module is shifted according to the action direction to infer the next grid state. Once we obtain the next grid code $\mathbf{g}_{t+1}$, its corresponding place code $\mathbf{p}_{t+1}$ is associated with the sensory input ($\mathbf{s}_{t+1}$).

In our implementation, we extend the grid cell modules to 2D space (with $\lambda_i^2$ dimensions for each one-hot grid cell module) and adapt the path integration described above to suit the proposed 2D sequential Morris Water Maze environments.

## 5  MESH FOR SPATIAL NAVIGATION (MSN)

### 5.1  MOTIVATION AND OVERVIEW

Artificial neural networks, despite their significant advancements, are still prone to a major short-coming known as 'catastrophic forgetting' during continual learning. This issue arises when these networks, after being trained on new tasks, tend to forget the old ones, thereby undermining their learning continuity. By contrast, natural organisms like rodents and humans showcase a remarkable resilience to such forgetting. This ability to continuously learn and adapt without forgetting past learning underscores the sophistication of biological learning systems. A wealth of scientific research has demonstrated that specific types of neurons, known as grid and place cells, play instrumental roles in counteracting catastrophic forgetting, particularly in the context of spatial memory. These cells, predominantly found in the hippocampus, are believed to create cognitive maps of the environment, helping the organism to navigate and remember spatial information.

Inspired by this, we design a novel method for continual learning based on MESH (Sharma et al., 2022) called MESH for Spatial Navigation (MSN). To begin with, the MESH converts observations into grid cell patterns (*grid code*). This involves representing the acquired data in a structured format that mimics the function of grid cells in the brain, which are integral to understanding spatial positioning and navigation. Next, the grid code is inputted into a randomly initialized fixed Convolutional Neural Network (CNN) to leverage its inherent spatial invariance, ensuring consistent output regardless of shifting inputs. Finally, an attention module takes the feature vector and retrieves the appropriate action based on features that have been observed previously.

### 5.2  ASSOCIATING GRID CODE DISPLACEMENTS WITH MOVEMENTS

We develop a model of how rodents rapidly learn to navigate in new environments. Using a randomly initialized fixed convolutional neural network (CNN), our model maps the rodent's current and goal locations (encoded in a grid code, provided by MESH) to a spatially-invariant representation of the *displacement* of the goal relative to its current position. We use an attention mechanism with the keys being the spatially-invariant representation of the grid code and the values being the appropriate actions. During the training phase, these key-value pairs are associated and stored within the mechanism. During the testing phase, the agent's current state generates a spatially invariant representation of displacement that is used as the query. This query is then processed through a dot

product operation with the existing keys in the dictionary. The action associated with the key most similar to our query is identified and used. This process allows for efficient action selection based on the spatially invariant displacement of the agent. Our architecture's spatial invariance allows the agent to rapidly learn to navigate in unseen environments *by only learning associations between new observations and the grid code* (it does not need to learn new associations to actions) as we will discuss in the next section. Figure 2 illustrates our model architecture. In our appendix Algorithm 1 illustrates the pseudocode of our model.

### 5.3 AGENT TRAINING AND ZERO-SHOT POLICY LEARNING IN NOVEL ENVIRONMENTS

The agent under consideration is now equipped with two crucial functionalities: the ability to counteract catastrophic forgetting and the capacity to facilitate forward transfer to novel environments. These two attributes together expedite the learning process.

During the initial phase of training, the agent is introduced to a novel environment where it initiates exploration. Concurrently, it collects observational data, forming associations between these observations and a grid code via the Memory Scaffold with Heteroassociation (MESH) framework. This process effectively constructs a memory scaffold, enabling the agent to effectively navigate within a specific environmental context.

Upon successful identification of the goal within the environment, the agent proceeds to store the corresponding grid code. This stored grid code, signifying the goal location, serves as a key reference point in the agent's cognitive map of the environment.

Subsequently, from multiple locations within the environment, we use our spatially invariant CNN to compute a representation of the vector displacement between the agent's location and the pre-stored goal location. This displacement vector encapsulates the navigational 'distance' the agent must traverse to reach the goal from its current position.

These displacement representations are then processed by the attention mechanism. The mechanism associates displacement with the corresponding movement action required to progress toward the goal. Storing these associations allows the agent to retrieve the appropriate action when a previously observed displacement is encountered later.

Upon introduction to a new environment, the agent embarks on a similar exploration phase. Once the goal is located in this new environment, a significant feature of our system emerges: *the policy requires no further training.* The agent first computes a representation of goal displacement using the spatially invariant CNN. Then, it uses applies the attention mechanism to the stored associations between displacement representations and movements to retrieve the correct navigational action.

This unique process facilitates zero-shot policy learning in new environments, underscoring the effectiveness and adaptability of our proposed framework. It exemplifies our agent's capacity to rapidly assimilate and apply knowledge, enabling successful navigation in unfamiliar environments.

## 6 EXPERIMENTS

### 6.1 EXPERIMENTAL DETAILS

We optimize parameters using Adam (Kingma & Ba, 2015) with a learning rate of $0.001$ for $800$ episodes for each environment. The maximum number of steps in each episode is set to $100$ and the starting configuration (head direction and coordinates) are different. The environment is a $30 \times 30$ grid with unique, noise-added step function markings on the walls. The agent has a field of view (FOV) of 120 degrees (see Figure 1a). We use a public continual learning implementation (Zhou et al., 2023) for EWC (Kirkpatrick et al., 2017) and implement our own version of replay buffer and fine-tuning. For fine-tuning, we sequentially train on each environment. In our replay buffer implementation, we allocated a fixed buffer size ($100$ in our case) during the training of the neural network within a single environment. Throughout this training phase, we stochastically selected data points for inclusion in our replay buffer. Upon completion of training in one environment, we initiated a fine-tuning process on our replay buffer by sampling from it, followed by an evaluation in all previously trained environments. This procedure was replicated across all five environments. We also examined the sensitivity to replay buffer size, as shown in the supplementary figure.

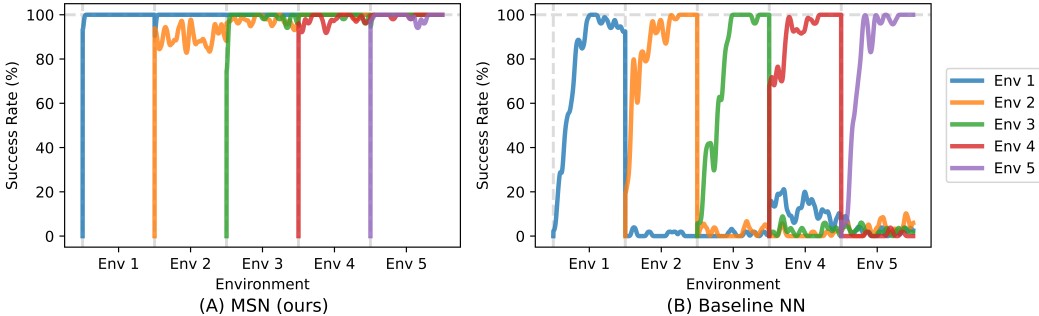

Figure 3: Our model avoids catastrophic forgetting. The average success rate of each environment while training on the following environments of our final model (a) and ours without MESH (b). The full model rapidly outperforms the one without MESH. In both plots, we use a moving average of 25 points and Gaussian smoothing with $\sigma = 10$.

Table 1: The average success rate (%) of each environment after training all environments. Our model maintains high success rates across the entire environment while other methods have bad performance except for the last environment (Env 5) due to catastrophic forgetting.

| Training Scheme | Average | Env 1 | Env 2 | Env 3 | Env 4 | Env 5 |
|---|---|---|---|---|---|---|
| Fine-Tune | 19.5 | 2.2 | 3.6 | 2.4 | 0.5 | 99.9 |
| EWC (Kirkpatrick et al., 2017) | 23.2 | 0.0 | 0.0 | 16.0 | 0.0 | **100.0** |
| Replay Buffer | 4.0 | 0.0 | 9.0 | 0.0 | 3.0 | 8.0 |
| Ours | **99.2** | **99.2** | **99.5** | **99.0** | **99.5** | 98.8 |

## 6.2 COMPARISON WITH BASELINES

In Figure 3 in Appendix, our approach (a) exhibits rapid learning in the first environment compared to the baseline neural network trained in a fine tuning framework shown in (b), where the observations are fed directly into a neural network and supervised by the correct action. Furthermore, our method successfully acquires a general, transferable navigation policy from this initial environment, allowing rapid navigation in subsequent environments *without any policy training*. This contributes to the prevention of catastrophic forgetting, as past environments can be recalled after recognizing the current environment through a few trajectories. In sharp contrast, the baseline experiments demonstrate an almost immediate onset of catastrophic forgetting upon exposure to a new environment. This phenomenon is marked by a rapid performance decline following the training of a few new trajectories, despite the initial successful knowledge transfer and adequate performance in the new setting.

To address this shortcoming of the baseline, we employed strategies additional strategies in continual learning such as the use of a replay buffer and Elastic Weight Consolidation (EWC) on the baseline neural network. Despite these efforts, both the replay buffer strategy and EWC demonstrated signs of catastrophic forgetting. Figure 4 displays the average success rate of all previously trained environments after training on the environment indicated on the x-axis. Our method consistently outperforms the continual learning baselines, whereas other methods exhibit degraded performance as more environments are introduced for training. Table 5 shows our method compared to baselines on all five environments after all training is complete.

The underwhelming performance of EWC in our tasks appears to stem from the similarity of inputs across different environments. Despite these similarities, the goal positions differ between environments. Consequently, similar observations could map to two distinct actions. EWC aims to maintain the weights of the network to find an overlap between all tasks. However, due to this subtle complexity in our task design, EWC fails to perform optimally.

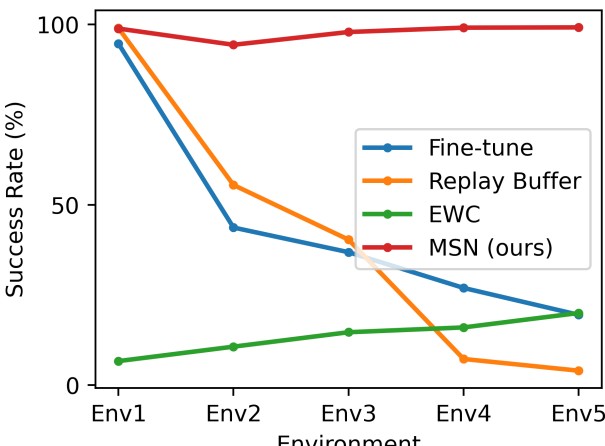

Figure 4: The average success rate along incremental stages. Our method clearly outperforms the continual learning baselines maintaining almost performance while other methods are degraded as training goes on.

## 6.3 ABLATION STUDY

The effectiveness of each individual component in our proposed method is analyzed and summarized in Table 2 in Appendix. Overall, the attention module plays a crucial role in achieving high performance. In fact, when used alone, the attention module achieves a perfect success rate. This is because when the goal location remains fixed, there is no need to rely on the spatial invariance provided by CNN (policy). Instead, the grid code can be directly associated with the attention module. This approach must learn associations between *every* observation and the corresponding ground-truth actions, which is memory inefficient and non-transferable to new environments. Furthermore, this approach becomes vulnerable when there are changes in the goal locations within the same environment since associations between observations and actions must be re-learned. On the other hand, when the attention module is combined with MESH without CNN, the performance is significantly lower. This is likely because MESH lacks spatial invariance, leading to the learning of conflicting associations between the grid code and actions. As for the CNN without the attention module, it corresponds to the "Fine-Tuning" model presented in Table 5. The use of MESH allows the CNN to use different input encoding methods, enhancing its versatility.

In summary, the superior accuracy demonstrated by the encoding network with attention, or by the attention mechanism in isolation, can primarily be attributed to its perfect memorization capabilities. This becomes apparent when the attention mechanism undergoes training as it is just storing key-value pairs. However, in the absence of such training, the model's performance in future environments significantly declines, often nearing zero. This reveals a lack of forward transfer or generalization capabilities in the model.

To verify the effectiveness of spatial invariance from CNN, we train one environment with the fixed goal location and evaluated it with the changed goal location. Table 3 shows that all three modules should be combined together to solve the new location with further training. Furthermore, we also test that using fully-connected layers (FC) instead of CNN cannot solve the problem, which emphasizes the need to spatial invariance to find unseen goal locations.

Our framework, which includes MESH, the encoding network, and the attention mechanism, is trained exclusively on one environment and subsequently evaluated on four *unseen* environments and one seen environment. Conversely, all other ablated models are trained and then evaluated in all five environments. We adopted this strategy due to the observation that, without any training in future environments, each of our ablation study networks merely exhibited random movement, demonstrating no ability to generalize.

Table 2: Combined components of MSN allows for zero-shot transfer to new environments. The last row is our final model. We measure the average success rate (%) across all environments after training the last environment. The exception is the case of MSN: MESH, the encoding network, and attention mechanism, where the system is trained in a single environment and subsequently tested across five different environments.

| MESH | Policy | Attention | Training Environment ID | Success Rate |
|---|---|---|---|---|
| | | ✓ | 1, 2, 3, 4, 5 | 100 |
| | ✓ | | 1, 2, 3, 4, 5 | 19.5 |
| | ✓ | ✓ | 1, 2, 3, 4, 5 | 99.7 |
| ✓ | ✓ | | 1, 2, 3, 4, 5 | 0.9 |
| ✓ | | ✓ | 1, 2, 3, 4, 5 | 5.4 |
| ✓ | ✓ | ✓ | 1 | 99.2 |

Table 3: Our model allows for adaptation to new goal locations not included during training. The last row is our final model, MSN. We measure the average success rate (%) of a new goal location after training the different goal in one environment. The exception is the case of MESH, the encoding network, and attention mechanism, where the system is trained in a single environment and subsequently tested across five different environments.

| MESH | Policy | Attention | Success Rate |
|---|---|---|---|
| | | ✓ | 1.6 |
| | ✓ | | 2 |
| | ✓ | ✓ | 1.8 |
| ✓ | ✓ | | 1 |
| ✓ | | ✓ | 5.6 |
| ✓ | ✓ (FC) | ✓ | 0.1 |
| ✓ | ✓ | ✓ | 99.5 |

## 7 DISCUSSION

We introduce a novel neural model, powered by the MESH architecture, which exhibits remarkable proficiency in rapidly learning and retaining knowledge across a range of environments. Furthermore, it facilitates an impressive transfer to unfamiliar settings. This capability for quick learning, generalization, and seamless adaptation represents a significant advancement in addressing complex cognitive tasks—tasks that often pose challenges to conventional machine learning methods but are effortlessly handled by biological agents.

Experimental results illuminate not only the successful application of structured neural models to complex real-world tasks but also the potential limitations of traditional deep learning methodologies. These methods have historically grappled with issues such as rapid learning, generalization, and the avoidance of catastrophic forgetting. In stark contrast, our model deftly navigates these hurdles, underscoring the potential benefits of incorporating inductive biases into neural models.

Our findings carry implications for both artificial intelligence research and neuroscience. They suggest a promising role for structured neural models, inspired by architectures found in the brain, in tackling complex tasks, thereby pushing the boundaries of what artificial intelligence systems can achieve. Given these encouraging results, we believe that continued exploration and development of structured neural models may herald significant advancements in the field. Looking ahead, it would be beneficial to explore how our proposed model could be further optimized or adapted to other tasks. Additionally, assessing its scalability and performance in even more complex, dynamic environments will be a valuable direction for future work.

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

# A    APPENDIX

## A.1    DYNAMIC SEQUENTIAL WATER MAZE

In this section, we extend our original task, developing a more complex paradigm called the Dynamic Sequential Water Maze (dsWM). In the previous model, an agent was positioned within five distinct environments, each containing a unique goal location. However, the increased complexity of dsWM necessitates a more advanced set of cognitive capabilities from the agent.

In this version of the task, the agent must now retain the goal position for each environment while additionally *adapting* to altered goal positions within a single environment. To elaborate, the agent is initially trained in one environment, with a fixed goal position. Following this training period, the goal position is changed twice, yet the agent's policy is not trained further. The agent is then tested a further 800 times each in two different goal positions. This procedure is subsequently repeated in four additional unique environments, without the initial training period.

In our primary study, we had compared our approach to several baseline models. These baseline models involved training the policy in one environment with a fixed goal position, followed by the relocation of the goal position within the same environment for testing. The results demonstrated that our method was unique in its ability to generalize without further training, while the baseline methods exhibited poor performance.

The Dynamic Sequential Morris Water Maze represents an extension of this original work, offering a more complex task and demanding greater cognitive adaptability from the agent. This enhanced task complexity will allow us to analyze the capability of our method further. Figure 6 shows that our MSN is robust against all inter- and intra-environment changes.

---

**Algorithm 1** Pseudocode for MSN

---

1: agent = Agent()                                                       ▷ Initialize agent
2: attention = Attention()                                          ▷ Initialize attention block
3: mesh = MESH($\Lambda$, $N_{\text{place cells}}$)       ▷ Initialize the mesh, $\Lambda$: set of grid periods for grid cells
4: all_obs = empty set()                                  ▷ Create an empty set to store all observations
5: **for** each env in envs **do**                                    ▷ Loop through each environment
6:     goal_state = null                                              ▷ Initialize goal as null
7:     **for** trial_number in $\mathsf{range}(n\_trajectories)$ **do**    ▷ Loop for a fixed number of trajectories
8:         **if** observation is associated with grid cell **then**
9:             found_where_i_am = True
10:         **else**
11:             found_where_i_am = False
12:         **end if**
13:         **for** each step in $\mathsf{range}(L)$ **do**                        ▷ $L$: max trajectory length
14:             obs = env.sensory_input()                     ▷ Get the current sensory input
15:             **if** trial_number = 0 **and not** found_where_i_am **then**
16:                 found_where_i_am = True
17:                 mesh.remap_grid(obs)                     ▷ Remap to a randomly set grid state
18:                 grid_state = mesh.get_grid_activations()
19:             **end if**
20:             **if** obs in all_obs and not found_where_i_am **then**
21:                 found_where_i_am = True
22:                 mesh.remap_grid(obs)                  ▷ Remap to set grid state to a visited location
23:                 grid_state = mesh.get_grid_activations()
24:             **end if**
25:             **if** found_where_i_am **then**
26:                 mesh.update_weights(obs)     ▷ Associate observation with the current grid state
27:                 all_obs.add(obs)
28:             **end if**
29:             **if** found_where_i_am and goal is not null **then**
30:                 displacement = $\mathsf{fixedCNN}$(goal_state, grid_state)          ▷ Calculate displacement
31:                 **if** in the first env **then**
32:                     action = Attention.associate(displacement, ground_truth_action)       ▷ Learn Association between displacement and action
33:                 **else**
34:                     action = Attention.retrieve(displacement)
35:                 **end if**
36:             **else**
37:                 action = random_action() ▷ Randomly wander until finding the goal and location
38:             **end if**
39:             agent.step(action)                                    ▷ Agent takes a step in the env
40:             mesh.update_grid(action)                  ▷ Update the grid state based on the action
41:             **if** env.reached_goal() **then**
42:                 goal_state = grid_state                     ▷ Update the goal if the agent reached it
43:                 **break**
44:             **end if**
45:         **end for**
46:     **end for**
47: **end for**

---

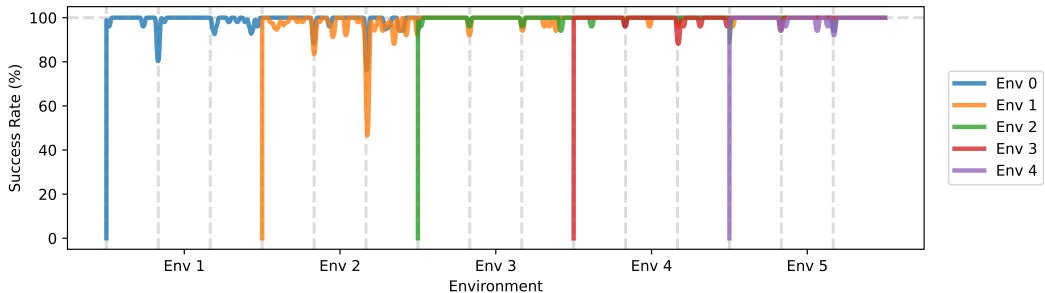

Figure 5: Our method shows robustness against all inter- and intra- environment changes. The average success rate of each environment while training on the following environments of our final model. Each dotted line indicates a goal position change. We use a moving average of 25 points and gaussian smoothing with $\sigma = 10$.

## A.2 REPLAY BUFFER

In order to assess the sensitivity of our baseline replay buffer, we conducted tests using a variety of replay buffer sizes, mirroring our original experimental setup. Initially, the network was trained in one environment and fine-tuned using 100 randomly sampled data points from the replay buffer. Subsequent testing was performed on all previously trained environments. The data obtained from these tests reveals no correlation between replay buffer size and performance in the latter environments, suggesting catastrophic forgetting.

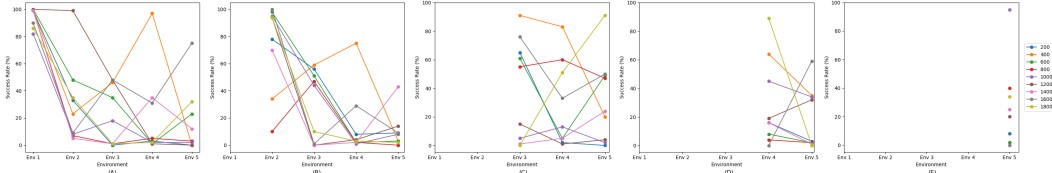

Figure 6: Each graph (A-E) depicts the model's performance accuracy in a specific environment (1-5). This accuracy is evaluated following initial training in a new environment and subsequent refinement via the replay buffer. The lines within each graph correspond to varying sizes of the replay buffer. Each data point on these lines represents the model's accuracy within the represented environment, post-training in the preceding environment.

## A.3 LARGE MORRIS WATER MAZE

To assess the adaptability of our system, we doubled the size of our Morris Water Maze and observed a minor performance degradation. This test was conducted using the same number of time steps as in the smaller maze configuration. Despite the reduction in search time relative to the increased area, the system still maintained a high level of performance.

Table 4: The average success rate (%) of each environment after training all environments.

| Training Scheme | Average | Env 1 | Env 2 | Env 3 | Env 4 | Env 5 |
|---|---|---|---|---|---|---|
| MSN (15 x 15) | **99.2** | **99.2** | **99.5** | **99.0** | **99.5** | **98.8** |
| MSN (30 x 30) | 93.1 | 94.5 | 94.25 | 87.25 | 95.19 | 94.4 |

## A.4 More Baselines

To benchmark against contemporary state-of-the-art methods, we selected four additional techniques: Dark Experience Replay (Buzzega et al., 2020), Dark Experience Replay ++ (Buzzega et al., 2020), A-GEM (Chaudhry et al., 2019), and Exerpeicen Replay (Rolnick et al., 2019).

In these benchmarks, we conducted tests over 100 epochs. For each epoch, the network is trained on 200 trajectories. Each trajectory was limited to a maximum of 100 time steps, after which we deemed it a timeout. Success was defined as the agent locating the goal within these 100 time steps. In methods utilizing a buffer, we set its capacity to 200. For the DER++ algorithm, we adhered to the optimal parameters recommended in the paper: $\alpha$ and $\beta$ both set at 0.5. All tested methods employed Cross Entropy loss and a learning rate of 0.001. Post-training in all five environments, we assessed the accuracy of each on the previous environments. As a point of comparison, we introduce our method wherein the action-selection network is trained exclusively in the first environment and then subjected to zero-shot testing in the remaining environments.

Table 5: The average success rate (%) of each environment after training all environments. Our model maintains high success rates across the entire environment while other methods have bad performance except for the last environment (Env 5) due to catastrophic forgetting.

| Training Scheme | Average | Env 1 | Env 2 | Env 3 | Env 4 | Env 5 |
|---|---|---|---|---|---|---|
| Fine-Tune | 19.5 | 2.2 | 3.6 | 2.4 | 0.5 | 99.9 |
| EWC (Kirkpatrick et al., 2017) | 23.2 | 0.0 | 0.0 | 16.0 | 0.0 | **100.0** |
| Replay Buffer | 4.0 | 0.0 | 9.0 | 0.0 | 3.0 | 8.0 |
| DER (Buzzega et al., 2020) | 2.05 | 0.0 | 7.69 | 2.56 | 0.0 | 0.0 |
| DER++ (Buzzega et al., 2020) | 3.39 | 0.0 | 3.85 | 0.0 | 11.54 | 1.54 |
| A-GEM (Chaudhry et al., 2019) | 7.31 | 0.0 | 7.69 | 15.38 | 13.46 | 0.0 |
| ER (Rolnick et al., 2019) | 16.92 | 23.08 | 26.92 | 30.77 | 3.85 | 0.0 |
| Ours(MSN) | **99.2** | **99.2** | **99.5** | **99.0** | **99.5** | 98.8 |

## A.5 Computing Infrastructure

We conducted our experiments on a high-performance computing system. The system was equipped with an AMD EPYC 7713 64-Core Processor, 32 GB of RAM and a Nvidia RTX 2080 ti GPU.

