# OpenReview forum: "Rapid Learning without Catastrophic Forgetting in the Morris Water Maze"
_ICLR.cc/2024/Conference — Submitted to ICLR 2024_

### Official Review · Reviewer_KBVt · 2023-10-29

**Soundness:** 2 fair
**Presentation:** 2 fair
**Contribution:** 1 poor
**Rating:** 3
**Confidence:** 3

**Summary:**

The paper studies the challenge of catastrophic forgetting in the context of continual learning. The proposed study focuses on the sequential Morris Water Maze (sWM) task which is inspired by several mechanisms used by biological systems. It combines a content-addressable memory system and a convolutional network architecture to implement these mechanisms in the context of ANNs. This model excels at fast learning, generalization, and continuous learning, outperforming baselines in both continuous and few-shot learning settings.

**Strengths:**

The paper reads well and can be followed straightforwardly.

**Weaknesses:**

1. It is not clear that the proposed task is of practical importance.


2. Comparison is extremely limited and considers old methods such as EWC.

3. The code is not provided which makes judgment about reproducibility challenging.

**Questions:**

The major novelty that I see in this paper is building connections between biological systems and ML. However, it is not clear how relevant the proposed task is and how effective it will be in the context of CL. The question is then why the paper has not done an evaluation according to the precedent given the fact that CL is an extremely well-established field by including SOTA methods.


============Post Rebuttal=============
Thanks for the rebuttal. I changed my rating accordingly. I don't find this work compelling and mostly find a proof-of-concept level work which is not clear whether it will be of practical relevance. The task is a limited synthetic task and I cannot think of a major benefit for future research.

---

> ### Author Response · Authors · 2023-11-23
>
> Thank you for your feedback on our manuscript. We will revise the manuscript, taking into account each of your valuable comments. We hope to address each of your questions and concerns here:
>
> > It is not clear that the proposed task is of practical importance.
>
> Our proposed water maze task is designed for testing neuroscience tasks using machine learning methods to reduce the gap between biological and artificial neural networks. We believe that this can give important insights into overcoming catastrophic forgetting, something biological agents do not suffer from. Our results indicate that, in contrast to standard continual learning methods which significantly struggle with catastrophic forgetting, our neuro-inspired method demonstrates substantially improved performance.
>
>
> >  Comparison is extremely limited and considers old methods such as EWC.
>
> Although EWC was published in 2016, it is still widely used due to its simplicity and fewer constraints (e.g., no need for replay buffer, multiple backbone) (Wołczyk et al. 2021, Wołczyk et al.  2022). We have also compared our method with other recent methods: DER, DER++, Experience Replay, A-GEM and included our results in the appendix Table 5.
>
> For convenience, here are the average success rates on all 5 environments after training on all 5, with the exception of our method, which we only trained our action-policy in the first environment and zero-shot test across the rest of the environments.
>
> | Training Scheme   | Average |
> |-------------------|---------|
> | Fine-Tune         | 19.5    |
> | EWC (Kirkpatrick et al., 2017) | 23.2    |
> | Replay Buffer     | 4.0     |
> | DER (Buzzega et al., 2020) | 2.05    |
> | DER++ (Buzzega et al., 2020) | 3.39    |
> | A-GEM (Chaudhry et al., 2019) | 7.31    |
> | ER (Rolnick et al., 2019) | 16.92   |
> | Ours (MSN)        | 99.2    |
>
>
>
> [1] Wołczyk, Maciej, et al. "Continual world: A robotic benchmark for continual reinforcement learning." Advances in Neural Information Processing Systems 34 (2021): 28496-28510.
>
> [2] Wolczyk, Maciej, et al. "Disentangling transfer in continual reinforcement learning." Advances in Neural Information Processing Systems 35 (2022): 6304-6317.
>
> > The code is not provided which makes judgment about reproducibility challenging.
>
>
> We attached the repository in the revision. https://anonymous.4open.science/r/water_maze-ED17/

---

### Official Review · Reviewer_3WUm · 2023-10-31

**Soundness:** 2 fair
**Presentation:** 1 poor
**Contribution:** 2 fair
**Rating:** 5
**Confidence:** 2

**Summary:**

The paper focuses on catastrophic forgetting in lifelong learning scenarios. Inspired by inspiration from the spatial learning mechanisms observed in biological neurons,  this paper introduces a new task, sequential Morris Water Maze (sWM), for rapid adaptation and continual learning.  Furthermore, the paper presents a lifelong learning approach built upon the Memory Scaffold with Heteroassociation (MESH) architecture, designed to promote generalization within the Water Maze environments.  However, the description of the proposed method requires further clarity.

**Strengths:**

The paper studies a practical and important problem: lifelong learning without catastrophic forgetting, focusing on the performance in sequential Morris Water Maze tasks.

The proposed method, a bio-inspired lifelong learning framework based on MESH, provides a reasonable method to mitigate catastrophic forgetting in changing environments.

Furthermore, in the experimental evaluation conducted on sequential Morris Water Maze tasks, the proposed method demonstrates superior performance compared to previous approaches, as reported in the paper.

**Weaknesses:**

1) Enhancing the paper's readability, especially for readers less familiar with neuroscience, would improve its significance. This could involve providing clearer (maybe more intuitive) explanations of certain concepts, such as the entorhinal cortex, the neocortical-entorhinal-hippocampal circuit, the memory scaffold, Hebbian learning, and grid cell patterns.

2) The description of the Morris Water Maze environment lacks clarity. What are the observations and sensory-cells in Figure 2. Is the observation meant to represent the agent's view as high dimensional vector? What are place and grid cells, and what are their dimensionalities? Does the memory store grid cells?  What are the differences between various environments, apart from variations in the goal positions?

3) The description of the proposed method requires further clarity. An explanation of the design architecture for the policy is needed. Additionally, specify which parts of the MESH incorporate attention mechanisms. How is the entire network trained? Is it trained using Reinforcement Learning? Define the objective function for training. Does the statement "The policy requires no further training" imply that the displacement network also doesn't require training?

4) The experimental evaluation lacks comprehensiveness. It would be valuable to compare the proposed method with more state-of-the-art continual learning approaches, such as ER [1] and A-GEN [2]. Moreover, in the paper [3], a strategy involving the replay of similar experiences is used for continual learning. It would be insightful to discuss the relationship between the proposed method and the paper [3] on memory replay.

[1] A. Chaudhry, et al “On tiny episodic memories in continual learning”, arxiv 2019.
[2] A. Chaudhry, et al “Efficient lifelong learning with a-gem”, ICLR 2018.
[3] A. Abulikemu, et al “Online Model Adaptation with Feedforward Compensation”, CoRL 2023.

**Questions:**

1) It is beneficial to provide clearer (maybe more intuitive) explanations of some concepts, such as neocortical-entorhinal-hippocampal circuit, memory scaffold, and grid cell patterns.

2) What are the observations and sensory cells in Figure 2? Does the memory store grid cells?  What are the differences between various environments, apart from variations in the goal positions?

3) How is the entire network trained? Is it trained through Reinforcement Learning?  What is the training objective?  Additionally, specify which part of the network utilizes attention mechanisms.

4) Consider highlighting the unique aspects of the proposed method compared to replay-based approaches, such as ER [1], A-GEM [2], and Feedforward [3].

---

> ### Author Response · Authors · 2023-11-23
>
> Thank you for your insightful and constructive feedback on our manuscript. We greatly appreciate the time and effort you have dedicated to reviewing our work. We will revise the manuscript, taking into account each of your valuable comments. We hope to address each of your questions and concerns here:
>
> >  It is beneficial to provide clearer (maybe more intuitive) explanations of some concepts, such as neocortical-entorhinal-hippocampal circuit, memory scaffold, and grid cell patterns.
>
>
> Memory scaffold is a memory structure with predefined fixed points that allow arbitrary sensory inputs to be remembered without catastrophic forgetting [1, 2].
>
> Grid cells are specialized cells in the entorhinal cortex that encode spatial information in a hexagonal pattern [3]. This cell is known for its crucial role in spatial navigation [4, 5].
>
> A neocortical-entorhinal-hippocampal circuit is a key neural circuit for spatial memory and navigation that combines three different components. The neocortex corresponds to sensory cells in MESH, the entorhinal cortex corresponds to grid cells and the hippocampal corresponds to place cells. Please refer to Sharma et al. (2022) [6] for details.
>
> [1] Mulders, D., et al.. A structured scaffold underlies activity in the hippocampus. bioRxiv, 2021.
>
> [2] Yim M. Y., et al., Place-cell capacity and volatility with grid-like inputs. eLife, 2021.
>
> [3] Moser EI, Moser MB. A metric for space. Hippocampus. 2008
>
> [4] Bush D, Barry C, Manson D, Burgess N. Using Grid Cells for Navigation. Neuron. 2015
>
> [5] Moser EI, Kropff E, Moser MB. Place cells, grid cells, and the brain's spatial representation system. Annu Rev Neurosci. 2008
>
> [6] Sharma S., et al. Content addressable memory without catastrophic forgetting
> by heteroassociation with a fixed scaffold. In ICML, 2022
>
>
>
>
>
>
>
>
>
> >  What are the observations and sensory cells in Figure 2? Does the memory store grid cells? What are the differences between various environments, apart from variations in the goal positions?
>
> The observations are the markings on the wall in the agent’s field of view, and they are passed into the sensory cells. The memory only stores the grid cell representation for the goal location once the agent reaches out there. The agent’s location is not stored.
> In each environment, the surrounding wall pattern is different as shown in Figure 1 (b). It leads to catastrophic forgetting of previous environments.
>
>
> > How is the entire network trained? Is it trained through Reinforcement Learning? What is the training objective? Additionally, specify which part of the network utilizes attention mechanisms.
>
> The entire network is trained through supervised learning. The observations and place cells are learned through hebbian-like association without backpropagation. The random projections from place cells to grid cells give us the grid cells. Please refer Sharma et al. (2022) for more details on associating observations and grid cells. The fixed convolutional neural network that takes the current grid code and the goal grid code and produces the invariant representation is not trained. This representation is then passed into the attention mechanism, and this is associated with the correct action through supervision. Essentially we have two networks, one for getting the grid code and another to get the correct action. The observation to grid code is learned in each environment, however the action network is only trained in the first environment, and zero-shot in the rest.
>
> > Consider highlighting the unique aspects of the proposed method compared to replay-based approaches, such as ER [1], A-GEM [2], and Feedforward [3].
>
> Our method does not rely on replaying previous trajectories as the methods mentioned. Using the high capacity of MESH, we are able to achieve high accuracy without having replay. In-addition, our method does not require any further training for the network mapping the grid-cell displacement to the action after training in the first environment, which is both more biological and more efficient. Other approaches such as the ones mentioned need training in every environment encountered.
>
> We have added 4 more modern methods for comparison in the appendix Table 5. We included DER, DER++, A-GEM, and Experience replay. For convenience, here are the average success rate on all 5 environments after training on all 5, with the exception of our method, which we only trained our action-policy in the first environment and zero-shot test across the rest of the environments.
>
> | Training Scheme   | Average |
> |-------------------|---------|
> | Fine-Tune         | 19.5    |
> | EWC (Kirkpatrick et al., 2017) | 23.2    |
> | Replay Buffer     | 4.0     |
> | DER (Buzzega et al., 2020) | 2.05    |
> | DER++ (Buzzega et al., 2020) | 3.39    |
> | A-GEM (Chaudhry et al., 2019) | 7.31    |
> | ER (Rolnick et al., 2019) | 16.92   |
> | Ours (MSN)        | 99.2    |

---

### Official Review · Reviewer_ndgr · 2023-10-31

**Soundness:** 3 good
**Presentation:** 4 excellent
**Contribution:** 2 fair
**Rating:** 6
**Confidence:** 3

**Summary:**

The paper introduces a novel continual learning benchmark based on the Morris Water Maze test of spatial learning in animals, as well as a dedicated neuroscience-inspired continual learning method combining Memory Scaffold with Heteroassociation framework, a randomly-initialised CNN, and an attention module. Through experiments on the sequential Morris Water Maze benchmark, the authors show their method outperforms standard continual learning baselines by effectively retaining past knowledge and quickly adapting to new environments.

**Strengths:**

The new benchmark is a valuable contribution to the continual learning community. The method has a strong neuroscientific grounding and it brings together existing components in an original way. The paper is well presented and nicely structured. The writing is clear and the figures are very helpful in conveying the main points of the argument. The ablation study provides sufficient justification for the individual design choices.

**Weaknesses:**

The empirical evaluation is the main weakness of the paper. The authors compare their method to only two continual learning baselines, both of which are quite old. In addition, the replay buffer sizes that are used in the experiments are rather small (200-1800) The proposed method seems to be custom-designed for the navigation task, so while it can serve as a model of how rodents learn to navigate in new environments, it is not a practical continual learning method that could be applied to an arbitrary task. To give existing baselines a fighting chance, I would recommend having two separate networks: one mapping observations and goal position into some latent representation and another mapping these to actions. In the first environment, train both networks. For each new environment, re-train only the first network.

**Questions:**

Why does replay buffer exhibit such poor performance on the last task?

Will the benchmark be made available? Is it a framework to produce random environments or just a fixed dataset?

Are the associations between displacement representations and actions simply memorised by the attention module?

Is the grid code of the goal location available to your method straight away?

How is the goal location provided to the network for replay and naive methods?

What exactly is stored in the rehearsal buffer? Observation-action pairs?

Have you tried increasing the grid size of the environment?

---

> ### Author Response · Authors · 2023-11-23
>
> Thank you for your insightful and constructive feedback on our manuscript. We greatly appreciate the time and effort you have dedicated to reviewing our work. We will revise the manuscript, taking into account each of your valuable comments. We hope to address each of your questions and concerns here:
>
> > The empirical evaluation is the main weakness of the paper. The authors compare their method to only two continual learning baselines, both of which are quite old. In addition, the replay buffer sizes that are used in the experiments are rather small (200-1800)
>
> While we don't disagree that the two baseline methods are a bit dated, EWC is still one of the main continual learning methods evaluated in the current literature [1, 2]. The replay buffer size we find is relatively proportionate to the environment, but as we show, increasing the replay buffer size doesn't have a significant difference.
>
> We have added 4 more modern methods for comparison in the appendix Table 5. We included DER, DER++, A-GEM, and Experience replay. For convenience here are the average success rate on all 5 environments after training on all 5, with the exception of our method, which we only trained our action-policy in the first environment and zero-shot test across the rest of the environments.
>
> | Training Scheme   | Average |
> |-------------------|---------|
> | Fine-Tune         | 19.5    |
> | EWC (Kirkpatrick et al., 2017) | 23.2    |
> | Replay Buffer     | 4.0     |
> | DER (Buzzega et al., 2020) | 2.05    |
> | DER++ (Buzzega et al., 2020) | 3.39    |
> | A-GEM (Chaudhry et al., 2019) | 7.31    |
> | ER (Rolnick et al., 2019) | 16.92   |
> | Ours (MSN)        | 99.2    |
>
>
>
> >  it is not a practical continual learning method that could be applied to an arbitrary task.
>
> Our method is specifically designed for spatial navigation tasks, rather than being a versatile continual learning tool applicable to any arbitrary task. The aim is to bridge the gap between natural and artificial neural networks, focusing on the specialized area of spatial navigation.
>
> >To give existing baselines a fighting chance, I would recommend having two separate networks: one mapping observations and goal position into some latent representation and another mapping these to actions. In the first environment, train both networks. For each new environment, re-train only the first network.
>
> In our method, we actually only train the association between the latent representation and action in one environment, this network is zero-shot tested in all other environments, where, exactly as you mention, the "first network" (finding the latent representation of the displacement between the goal and current position) is the only one re-trained in multiple environments.
>
>
> >Why does the replay buffer exhibit such poor performance on the last task?
>
> Unlike other baselines, a model with a replay buffer directly uses trajectories of previous environments during training. Each environment has a different goal position and background with the same action space (output space). Therefore, training with previous environments interferes the performance degradation. Moreover, we believe that this is the main reason why other baselines suffer from huge catastrophic forgetting, unlike standard class incremental learning.
>
> >  In addition, the replay buffer sizes that are used in the experiments are rather small (200-1800)
>
> The replay buffer size is usually set to 2,000 in iCIFAR10, iCIFAR100, and ImageNet100 for 5 or 10 tasks [1, 2, 3]. This is almost similar to the maximum size of the replay buffer we showed in Figure 6 as the reviewer noticed. Appendix A.2 shows that the larger memory size does not improve the performance.
>
> [1] Rebuffi, Sylvestre-Alvise, et al. "icarl: Incremental classifier and representation learning." CVPR. 2017.
>
> [2] Yan, Shipeng, Jiangwei Xie, and Xuming He. "Der: Dynamically expandable representation for class incremental learning." CVPR. 2021.
>
> [3] Zhou, Da-Wei, et al. "A model or 603 exemplars: Towards memory-efficient class-incremental learning." ICLR 2023.
>
>
> > Are the associations between displacement representations and actions simply memorized by the attention module?
>
> Yes, the attention model memorizes the associations between displacement representations and actions.
>
> > Is the grid code of the goal location available to your method straight away?
>
> No, the grid code is not available initially. There is a period of exploration in the environment during which the agent orients itself by associating sensory observations with grid states. After this period, the grid code becomes available. In future trials in an environment, the grid code only is "available" when the agent learns where it is (i.e. it localizes itself in the environment).
>
> > What exactly is stored in the rehearsal buffer? Observation-action pairs?
>
> The replay buffer stores exploration trajectories (observation-action pairs) sampled from training on each task.

---

> > ### Author Response · Authors · 2023-11-23
> >
> > > Have you tried increasing the grid size of the environment?
> >
> > Yes, we chose this size of environment because even at a relatively small size, deep learning methods, even when equipped with continual learning methods, fail. We have included the appendix Table 4 a larger grid size and the performance of our network in that larger size. Performance drops a little, however we are using the same time steps as the smaller network, thus a shorter time relative to the search space.
> >
> > > Will the benchmark be made available? Is it a framework to produce random environments or just a fixed dataset?
> >
> > Yes! We will release our codebase with the benchmark. This is a framework to produce random environments and can be varied in difficulty by modifying the size of the environment.
> > https://anonymous.4open.science/r/water_maze-ED17/

---

### Official Review · Reviewer_h9UH · 2023-11-04

**Soundness:** 3 good
**Presentation:** 3 good
**Contribution:** 3 good
**Rating:** 6
**Confidence:** 4

**Summary:**

This paper proposes a task-specific maze path-finding network that is suitable for continual learning. The key contribution is to decouple the policy, and the localization module, and the memorized goal location. The experiments show that the proposed network can learn 5 environments with no forgetting, whereas the baseline policy network completely forgets the previous task.

**Strengths:**

- The idea of an end-to-end network that is capable of continual learning is interesting, even if it can only handle path finding tasks.
- The results suggest that the proposed network clearly solves continual learning.

**Weaknesses:**

- It makes sense that the policy network is invariant across tasks, since given true localization and goal location it only needs to learn a good search algorithm. But the place cell and grid cell may still suffer from catastrophic forgetting. Does the model get another set of newly initialized place cells and grid cells when switching to a different environment? Otherwise, how does it prevent forgetting? I would appreciate further clarification on this part.
- I understand that the proposed method is tailored to a path finding task, however, to make it more generalizable, it would be better to test on other types of maze tasks (perhaps with more complex visual features and map topology). Moreover, I don’t see why the task needs to be a water maze (with no walls) vs. a real maze.
- I would appreciate more clarity on model training and loss functions. An algorithm block can also strengthen the presentation clarity of the paper.

**Questions:**

See above.

---

> ### Author Response · Authors · 2023-11-23
>
> Thank you for your insightful and constructive feedback on our manuscript. We greatly appreciate the time and effort you have dedicated to reviewing our work. We will revise the manuscript, taking into account each of your valuable comments. We hope to address each of your questions here:
>
> > It makes sense that the policy network is invariant across tasks, since given true localization and goal location it only needs to learn a good search algorithm. But the place cell and grid cell may still suffer from catastrophic forgetting. Does the model get another set of newly initialized place cells and grid cells when switching to a different environment? Otherwise, how does it prevent forgetting? I would appreciate further clarification on this part.
>
> No, place cells and grid cells are not initialized for each environment. When initializing the agent, we initialize one set of place cells and grid cells that the agent uses for all environments. The high capacity of the MESH network allows for multiple environments, and when the agent is placed in a new environment, only the associations between the sensory cells and places cells are learned through Hebbian-like associations. The projections from the grid cell and place cell are predefined random projections that are initialized once during the initialization of the agent. Critically, this setup *prevents place cells and grid cells from catastrophic forgetting*. Even as more and more sensory-grid cell associations are learned, the network retains separate grid and place representations for distinct sensory inputs: it avoids interference.
>
> > I understand that the proposed method is tailored to a path-finding task, however, to make it more generalizable, it would be better to test on other types of maze tasks (perhaps with more complex visual features and map topology). Moreover, I don’t see why the task needs to be a water maze (with no walls) vs. a real maze.
>
> We hope to test more types of maze tasks in the future, but we chose Morris WaterMaze as an example for 2 primary reasons. First, this is a task that standard deep learning and continual learning methods struggle with as we show experimentally. Second, this task is a popular neuroscience task that remains to be modeled. [1, 2] On this neuroscience task, we find that a structured neural model inspired by biological circuits can outperform standard deep learning. This task is also difficult in the sense that the agent must localize based solely on the markings of the walls: the agent receives no additional location-related signals. Note that the observation is not a bird-eye view of the 2D map.
>
> > I would appreciate more clarity on model training and loss functions. An algorithm block can also strengthen the presentation clarity of the paper.
>
> We have added a new algorithm block in the appendix. We do not use any gradient descent in our method, rather we are using heteroassociation between sensory input and grid cell representation. For the baselines, we use cross entropy loss with the ground truth action.
>
> [1] Vorhees C, Williams M. T. Morris water maze: procedures for assessing spatial and related forms of learning and memory. Nat Protoc. 2006
>
> [2] Redish AD, Touretzky DS. The role of the hippocampus in solving the Morris water maze. Neural Comput. 1998

---

### Meta-Review · Area_Chair_ey37 · 2023-12-08

**Metareview:**

This paper studies the plasticity-stability dilemma in Continual Learning in the context of a new task, the Morris Water Maze. Taking inspiration from biology, the authors use a memory system to retain knowledge and a spatially invariant convolutional network architecture for rapid adaptation.

In terms of strengths, the reviewers commented on the benchmark being a valuable contribution (ndgr), the bringing together of ideas in an interesting (ndgr, 3WUm) and effective way in solving the CL task of interest (h9UH, 3WUm), as well as the paper being well structured and clearly written (KBVt, ndgr), although some concerns exist regarding terminology difficult to access for readers not familiar with the biological inspiration (3WUm).

Concerns exist primarily in the method being designed “specifically designed for spatial navigation tasks, rather than being a versatile continual learning tool applicable to any arbitrary task” (authors), making the empirical evaluation against general methods somewhat unsurprising and perhaps less convincing. In the authors’ defense, it should be recognized that a representative range of additional baselines was added during the rebuttal against which the presented method compares favorably.

My main hesitation with this submission is that ICLR (and its CL community) might not be the best venue for this submission and that a journal or conference more targeted at interdisciplinary work would both be able to provide more feedback on the biological inspiration as well as show more enthusiasm for this submission. As it stands, the submission is unfortunately narrowly below the acceptance threshold, primarily because of the open question about relevance and impact.

**Justification For Why Not Higher Score:**

See Meta-Review.

**Justification For Why Not Lower Score:**

N/A

---

### Decision · Program_Chairs · 2024-01-16

Reject